



# Seamless streamflow model provides forecasts at all scales from daily to monthly and matches the performance of non-seamless monthly model

David McInerney[1], Mark Thyer[1], Dmitri Kavetski[1], Richard Laugesen[2], Fitsum Woldemeskel[3], Narendra Tuteja[2], George Kuczera[4]

[1]School of Civil, Environmental and Mining Engineering, University of Adelaide, SA, Australia
[2]Bureau of Meteorology, ACT, Canberra, Australia
[3]Bureau of Meteorology, VIC, Melbourne, Australia
[4]School of Engineering, University of Newcastle, Callaghan, NSW, Australia

*Correspondence to*: David McInerney (david.mcinerney@adelaide.edu.au)

**Abstract.** Subseasonal streamflow forecasts inform a multitude of water management decisions, from early flood warning to reservoir operation. 'Seamless' forecasts, i.e., forecasts that are reliable over a range of lead times (1-30 days) and when aggregated to multiples time scales (e.g. daily and monthly) are of clear practical interest. However, existing forecasting products are often 'non-seamless', i.e., designed for a single time scale and lead time (e.g. 1 month ahead). If seamless forecasts

are to be a viable replacement for existing 'non-seamless' forecasts, it is important that they offer (at least) similar predictive performance at the time scale of the non-seamless forecast.

This study compares the recently developed seamless daily Multi-Temporal Hydrological Residual Error (MuTHRE) model to the (non-seamless) monthly streamflow post-processing (QPP) model that was used in the Australian Bureau of Meteorology's Dynamic Forecasting System. Streamflow forecasts from both models are generated for 11 Australian

catchments, using the GR4J hydrological model and post-processed rainfall forecasts from the ACCESS-S climate model. Evaluating monthly forecasts with key performance metrics (reliability, sharpness, bias and CRPS skill score), we find that the seamless MuTHRE model provides essentially the same performance as the non-seamless monthly QPP model for the vast majority of metrics and temporal stratifications (months and years). When this outcome is combined with the numerous practical benefits of seamless forecasts it is clear that seamless forecasting technologies, such as the MuTHRE model, are not

only viable, but a preferred choice for future research development and practical adoption of streamflow forecasting.





## 1. Introduction

Subseasonal streamflow forecasts (with lead times up to 30 days) can be used to inform a range of water management decisions,
from flood warning and reservoir flood management at shorter lead times (e.g. up to a week) to river basin management at
time scales up to a month. Since different applications require forecasts over a range of lead times and time scales, recent
research has focussed on producing *seamless* forecasts, i.e. forecasts from a single product that are reliable and sharp across
multiple lead times and aggregation time scales (McInerney et al., 2020). Current forecasting practice often employs more
traditional *non-seamless* forecasts, i.e. forecasts that are developed and applicable at only a single lead time and time scale
(e.g., Mendoza et al., 2017; Gibbs et al., 2018; Woldemeskel et al., 2018). For seamless forecasts to be a viable replacement
for *non-seamless* forecasts, it is important to establish that the performance of seamless forecasts is competitive with their non-
seamless counterparts at the native time scale of the latter. This is the focus of our study.

Recent research by McInerney et al. (2020) has shown that seamless subseasonal forecasting is achievable. That study
developed the Multi-Temporal Hydrological Residual Error (MuTHRE) model, which represents seasonality, dynamic biases
and non-Gaussian errors. Using a case study with 11 catchments in the Murray Darling Basin, Australia, it was concluded that
subseasonal forecasts generated using the MuTHRE model are indeed *seamless:* daily forecasts are consistently reliable (i) for
lead times between 1 and 30 days, and (ii) when aggregated to monthly forecasts.

Seamless subseasonal forecasts, from residual error models such as MuTHRE, produce reliable forecasts over a wide range of
aggregation time scales (e.g. daily to monthly) and lead times (1-30 days). In contrast, non-seamless forecasts are only available
at a single time scale (e.g. monthly), and cannot be aggregated to longer time scales. The practical benefits of this are outlined
as follows:

1. **Seamless forecasts can be used to inform decisions at a range of time scales.** Forecast *users* can utilize seamless
   subseasonal forecasts to inform a wide range of decisions, including

   - Flood warning, where short-term forecasts (up to 1 week) on individual days are of practical interest (Cloke and
Pappenberger, 2009);

   - Hydro-electric reservoir management, which can utilize forecasts of inflow between 7 and 15 days to increase
     production in the electricity grid (Boucher and Ramos, 2019);

   - Managing reservoirs for rural water supply, where forecast volumes over long aggregation scales (e.g. weeks/months),
     and at long lead times (up to 1 month), are required due to long travel times (Murray-Darling Basin Authority, 2019);

- Operation of urban water supply systems, where monthly forecasts are of value (Zhao and Zhao, 2014).

2. **Seamless daily forecasts are easily integrated into river system models used for real-time decision-making.** Perhaps
   the greatest potential for seamless forecasts is their use as input into real-time decision-making tools used by urban and
   rural water authorities. These tools include river system models (e.g. eWater Source, Welsh et al., 2013), which run natively
   at the daily scale and are used to inform resource management decisions over larger time scales. Streamflow forecasts from
non-seamless models cannot be used as input into these models, since they do not match the required input time scale of
   the river system model, and are not reliable when aggregated to longer time scales (e.g. from daily to monthly).





3. **Seamless forecasts simplify forecasting systems, as a single seamless product can serve a range of forecast requirements at different time scales.** As forecasts are often required at multiple time scales (daily-monthly), non-seamless forecast strategies require developing multiple post-processing models at specific time scales of interest (e.g. a daily model and a monthly model). Seamless forecasts offer practical benefits to *forecast-providers*, such as the Australian Bureau of Meteorology, as they reduce the need to develop multiple non-seamless forecasts for different applications. A seamless forecasting system offers a single product that can serve a wide range of forecast requirements.

These practical benefits of seamless forecasts provide a clear motivation for their development and use. However, for seamless forecasts to be a viable replacement for non-seamless forecasts, it is important that they do not come at the cost of a substantial drop in performance at the native time scale of the non-seamless forecast. For example, if aggregated forecasts from a seamless daily model were considerably worse than monthly forecasts from an existing non-seamless model, users of the monthly forecasts would prefer to continue using forecasts from the non-seamless model. In general, one might expect a non-seamless model, which is developed and calibrated at single time scale, to provide superior performance than seamless forecasts calibrated at shorter time scale and then aggregated. While the non-seamless model has only one job to do, which is to provide quality forecasts at single time scale, the seamless model is expected to produce good performance over a range of lead times and aggregation time scales. Herein lies a major challenge of seamless forecasting.

Our interest in comparing the performance of aggregated seamless forecasts with non-seamless forecasts at their native time scale has similarities to previous research in aggregating deterministic streamflow predictions. For example, Wang et al. (2011) found that the WAPABA monthly rainfall-runoff model produced similar/better performance than aggregated predictions from the SIMHYD/AWBM daily rainfall-runoff models, despite only using observed monthly forcing data. Yang et al. (2016) compared daily and sub-daily versions of the SWAT model (with daily and sub-daily observed rainfall inputs) and found large differences in the partitioning of baseflow and direct runoff. However, to the best of the authors' knowledge, no studies have compared aggregated *probabilistic* forecasts from a seamless model against probabilistic forecasts from a non-seamless model. The aim of this study is to *establish whether aggregated forecasts from a seamless model achieve comparable performance to those from a non-seamless forecasting model at its native time scale*. This aim is achieved by comparing the monthly forecast performance of the seamless MuTHRE model (aggregated from daily to monthly) against the non-seamless monthly streamflow post-processing model of Woldemeskel et al. (2018), used in the Australian Bureau of Meteorology's Dynamic Forecasting System.

The remainder of the paper is organized as follows. Section 2 describes the QPP models and Section 3 introduces the case study methods used to evaluate the QPP models. Sections 4 and 5 present and discuss results, while Section 6 provides concluding remarks.


## 2. Forecasting models

This section describes the seamless MuTHRE daily streamflow post-processing (QPP) model and the non-seamless monthly

QPP model.

### 2.1. Probability model

Both QPP models can be represented as a probability model ($Q_t$) for streamflow $q_t$ at time $t$,

$$q_t \sim Q_t\left(\boldsymbol{\theta}; \mathbf{x}_t, \mathbf{s}_{t-1}, \tilde{\mathbf{q}}_{\boldsymbol{\omega}(t_0)}\right) \tag{1}$$

where $\boldsymbol{\theta}$ are parameters of the hydrological and error models (described below), $\mathbf{x}_t$ are inputs to the hydrological model,

including rainfall, $\mathbf{s}_{t-1}$ are states of the hydrological model at time $t-1$, and $\tilde{\mathbf{q}}_{\boldsymbol{\omega}(t_0)}$ are observed streamflow from a

preceding time period $\boldsymbol{\omega}(t_0)$ up to time $t_0$. The units for time $t$ are days for the seamless MuTHRE model, and months for

the non-seamless monthly QPP model.

The probability model $Q_t$ is a residual error model, where a residual error $\eta_t$ is added (in transform space) to a deterministic

prediction $q_t^{\text{det}}$,

$$z(Q_t; \boldsymbol{\theta}_z) = z(q_t^{\text{det}}; \boldsymbol{\theta}_z) + \eta_t \tag{2}$$

The deterministic prediction $q_t^{\text{det}}$ is computed from a deterministic hydrological model $h(\boldsymbol{\theta}_h; \mathbf{x}_t, \mathbf{s}_{t-1})$, as detailed in

Sections 2.2.2 and 2.3.2 for the seamless and non-seamless approaches respectively. Note that the hydrological model $h$ has

parameters $\boldsymbol{\theta}_h$, which for the purposes of describing the QPP model are assumed to be pre-calibrated.

The model $h$ is used to produce

i.    "Simulated" streamflow $\mathbf{q}^{\text{sim}}$, when observed rainfall $\tilde{\mathbf{x}}$ is used as input forcing in $h$, and

ii.    An ensemble of "raw" streamflow forecast replicates, $\{\mathbf{q}^{\text{raw}(f)}; f=1,\ldots,N_{\text{foc}}\}$, when an ensemble of $N_{\text{foc}}$ forecast

rainfall replicates $\{\mathbf{x}^{\text{foc}(f)}; f=1,\ldots,N_{\text{foc}}\}$ is used to force $h$. The raw streamflow forecasts account for forecast rainfall

uncertainty, but do not account for hydrological uncertainty associated with errors in hydrological model structure and

initial conditions.

The transformation $z$, with parameters $\boldsymbol{\theta}_z$, is used to reduce both heteroscedasticity and skewness in residuals. We choose

the Box Cox transformation (e.g., Box and Cox, 1964),





$$z(q;\lambda,A) = \begin{cases} \dfrac{(q+A)^{\lambda}-1}{\lambda} & \text{if } \lambda \neq 0 \\ \log(q+A) & \text{otherwise} \end{cases} \tag{3}$$

with parameters $\boldsymbol{\theta}_z = \{\lambda, A\}$. The power parameter $\lambda$ is set to 0.2 in both QPP models (McInerney et al., 2017). For the seamless MuTHRE model, the offset parameter $A$ is inferred as part of the hydrological model calibration (McInerney et al., 2020), while for the non-seamless monthly QPP model it is set to 1% of the mean observed monthly streamflow, i.e. $A = 0.01 \times \text{mean}(\tilde{\mathbf{q}}^{\text{mon}})$ (Woldemeskel et al., 2018).

The residual error term $\eta_t$ is modelled as an AR(1) process after standardization,

$$v_t = \phi_\eta v_{t-1} + y_t \tag{4}$$

$$v_t = (\eta_t - \mu_t)/s_t \tag{5}$$

where $\mu_t$ and $s_t$ are the (time-varying) mean and scaling factor of $\eta_t$, $\phi_\eta$ is the lag-1 autoregressive parameter, and $y_t$ is the random component (referred to as the "innovation") at time $t$.

When generating forecasts, recent streamflow observations are used to update errors via the AR(1) model, and reduce uncertainty in $\eta_t$ for short lead times.

**2.2. Seamless MuTHRE model**

**2.2.1. Overall approach**

The seamless MuTHRE model operates at the daily time scale.

Uncertainty due to both forecast rainfall and hydrological errors is represented using the ensemble dressing approach (Pagano et al., 2013). The ensemble of daily raw streamflow forecasts, $\mathbf{q}^{\text{raw}}$, obtained by propagating an ensemble of rainfall forecasts through the hydrological model $h$ (as described in Section 2.1), accounts for forecast rainfall uncertainty. A randomly generated replicate of the residual term, $\boldsymbol{\eta}$, is then added to each replicate of the raw streamflow forecasts to account for hydrological uncertainty. Note that this approach to capturing forecast rainfall and hydrological uncertainty relies on the rainfall forecasts being reliable in order to produce reliable streamflow forecasts (Verkade et al., 2017).

**2.2.2. Deterministic model implementation**

In the context of equation (2), the deterministic term in the MuTHRE model at its *daily* time step $t$ is

$$q_t^{\text{det}} = q_t^{\text{raw}(f)} = h(\boldsymbol{\theta}_h; \mathbf{x}_t^{(f)}, \mathbf{s}_{t-1}) \tag{6}$$

i.e., the residual error model is applied to each individual raw forecast.



### 2.2.3. Residual error model implementation

The MuTHRE model assumes that the mean of the residual error – $\mu_t$ in equation (5) – varies in time due to "seasonality"

and "dynamic biases" (associated with hydrologic non-stationarity),

$$\mu_t = \mu_{d(t)}^{(s)} + \mu_t^{(b)} + \mu^* \tag{7}$$

The seasonality component $\mu_{d(t)}^{(s)}$ describes the mean value of $\boldsymbol{\mu}$ on the day-of-the-year $d(t)$, the dynamic bias term $\mu_t^{(b)}$

describes the mean value of $\boldsymbol{\mu}$ (after removing seasonality) over the preceding $N_b$ days ( $N_b = 30$ is used), and $\mu^*$ is a

remaining constant. Full details of these terms are provided in McInerney et al. (2020).

In the MuTHRE model, the scaling factor – $s_t$ in equation (5) – is constant (set to 1 for simplicity).

Innovations are modelled using a two-component mixed-Gaussian distribution

$$y_t \sim \mathcal{N}_{\text{mix}}\left(0, \sigma_1^2, 0, \sigma_2^2, w_1\right) \tag{8}$$

where $\sigma_1$ and $\sigma_2$ are the standard deviations of the two components, and $w_1$ is the weight of the first component (with

component means set to zero). Compared to a standard Gaussian distribution, the mixed-Gaussian distribution allows for fatter

tails (i.e., excess kurtosis) in the distribution of innovations, which has been shown to improve reliability of daily forecasts at

short lead times (Li et al., 2016; McInerney et al., 2020).

### 2.2.4. Calibration

In the seamless MuTHRE model the residual term $\boldsymbol{\eta}$ represents hydrological uncertainty only, i.e. it does not include forecast

rainfall uncertainty. The parameters of the residual error model $\{\phi_\eta, \sigma_1^2, \sigma_2^2, w_1\}$ are calibrated using data at the *daily* scale:

(i) Daily hydrological model simulations forced with observed rainfall ( $\mathbf{q}^{\text{sim}}$, described in Section 2.1)

(ii) Daily observed streamflow ( $\tilde{\mathbf{q}}$ ).

Full details of the calibration procedure are provided in McInerney et al. (2020).

### 2.3. Non-seamless monthly QPP model

### 2.3.1. Overall approach

The non-seamless monthly QPP model operates at the monthly time scale. The ensemble of daily raw streamflow forecasts

$\{\mathbf{q}^{\text{raw}(f)}; f = 1, \ldots, N_{\text{foc}}\}$ (obtained using the rainfall ensemble as input to the hydrological model $h$) is aggregated from the

daily to monthly time scale $\{\mathbf{q}^{\text{raw,mon}(f)}; f = 1, \ldots, N_{\text{foc}}\}$, and then collapsed to a deterministic forecast by taking the median





value at each time step $\mathbf{q}^{\text{det,mon}}$ (i.e. the uncertainty from the raw streamflow replicates is discarded). Combined forecast rainfall and hydrological uncertainty is then represented through the residual term $\boldsymbol{\eta}$, with replicates of $\boldsymbol{\eta}$ added to the

deterministic forecast $\mathbf{q}^{\text{det,mon}}$ to produce the monthly streamflow forecasts.

### 2.3.2.    Deterministic model implementation

In the context of equation (2), the deterministic term in the non-seamless model at its *monthly* time step $t$ is computed as follows,

$$q_t^{\text{raw,mon}(f)} = \text{average}\left(q_{t*}^{\text{raw}(f)}; t* \in T(t)\right) \tag{9}$$


$$q_t^{\text{det}} = \text{median}\left(q_t^{\text{raw,mon}(f)}; f = 1,\ldots,N_{\text{foc}}\right) \tag{10}$$

where $T(t)$ is averaging window (range of days) corresponding to the monthly time step $t$. In other words, the residual error model is applied at the monthly scale and after collapsing the ensemble of raw forecasts to a single time series.

### 2.3.3.    Residual error model implementation

The residual error model captures seasonality in residuals by varying the mean $\mu_t$ and scaling factor $s_t$ by month. Innovations are assumed to be independent and identically distributed Gaussian,

$$y_t \sim \mathcal{N}\left(0,\sigma_y^2\right) \tag{11}$$

where $\sigma_y$ is the standard deviation of the innovations.

### 2.3.4.    Calibration

In the non-seamless monthly QPP model the residual term $\boldsymbol{\eta}$ represents combined forecast rainfall and hydrological uncertainty. The parameters of the residual error model $\left\{\phi_\eta, \sigma_y^2, \{\mu_m, s_m; m = 1,\ldots,12\}\right\}$ are calibrated using data at the *monthly* scale:

(i)    Monthly deterministic forecasts obtained using forecast rainfall ($\mathbf{q}^{\text{det,mon}}$, described in Section 2.3.1)

(ii)    Monthly observed streamflow ($\tilde{\mathbf{q}}^{\text{mon}}$)

Full details of the calibration approach are provided in Woldemeskel et al. (2018).

### 2.4.    Differences between the MuTHRE and monthly QPP models

The seamless MuTHRE and non-seamless monthly QPP models differ in their model structure, and hence their approach to calibration and forecasting.



### 2.4.1. Differences in model structure

The residual error models used in the MuTHRE and monthly QPP models represent different sources of uncertainty and have differences in their implementations, as outlined below and shown in Figure 1a:

- The seamless MuTHRE model uses a *daily* residual error model to capture *only hydrological uncertainty*. It applies residual errors directly to raw streamflow forecasts

- The non-seamless monthly QPP model uses a *monthly* residual error model to capture *both hydrological and rainfall*
*uncertainty*. Daily raw streamflow forecasts are aggregated to monthly raw streamflow forecasts, then the ensemble is reduced down to its median, and residual errors are applied to this monthly time series.

The residual error model used in the seamless MuTHRE model and non-seamless monthly QPP models differ in their implementation of the following aspects:

a)   *Seasonality*. The seamless MuTHRE model only varies the mean with the time of year, while the non-seamless monthly
QPP model varies both the mean and scaling factor.

b)   *Incorporation of recent streamflow data*. The seamless MuTHRE model uses monthly observed streamflow from the previous month to update recent hydrological biases, as well as daily observed streamflow from the most recent day to update the daily AR(1) model. The non-seamless monthly QPP model only uses monthly observed streamflow from the previous month to update the monthly AR(1) model.

c)   *Distribution used to model innovations*. The seamless MuTHRE model uses a mixed-Gaussian distribution for daily innovations, compared with the non-seamless monthly QPP model which uses a standard Gaussian distribution for monthly innovations. The mixed-Gaussian distribution has been shown to improve reliability of daily forecasts at short lead times, but does not impact performance of daily forecasts at longer lead times, or when aggregated to the monthly scale (McInerney et al., 2020).

### 2.4.2. Differences in calibration approach

Both approaches use the same deterministic hydrological model calibrated using observed rainfall and streamflow data. However, due to structural differences in their representation of residual errors, the seamless MuTHRE and non-seamless monthly QPP models differ in approach used to calibrate the residual error model parameters. The key differences are illustrated in Figure 1(b) and outlined below:

- The seamless MuTHRE model uses hydrological model simulations forced by *observed* rainfall, while the non-seamless monthly QPP model parameters uses *forecast* rainfall as input to the hydrological model;

- The seamless MuTHRE model is calibrated using *daily* observed streamflow data, while the non-seamless monthly QPP model is calibrated using *monthly* observed streamflow.






**Figure 1: Conceptual diagrams illustrating key differences between the seamless MuTHRE model and non-seamless monthly QPP model.** The diagram for model structure in panel (a) shows the MuTHRE model applies daily hydrological errors to raw streamflow forecasts to produce an ensemble of daily streamflow forecasts, while the monthly QPP model aggregates raw forecasts to the monthly scale, collapses the ensemble to its median value, and applies a monthly residual error model to accounts for both hydrological and forecast rainfall uncertainty. The diagram for the calibration approach in panel (b) shows key differences in how the models are calibrated (highlighted in colored text) are (1) the MuTHRE model is forced by *observed* rainfall during calibration, while the monthly QPP model is forced with *forecast* rainfall, (2) the MuTHRE model is calibrated using observed *daily* streamflow data, while the monthly QPP model uses observed *monthly* streamflow. The diagram for forecasting in panel (c) shows that the MuTHRE model produces seamless daily streamflow forecasts, which can be used at a *range of lead times and aggregation periods (e.g. daily, weekly, fortnightly, monthly)*, while the monthly QPP model produces only *monthly forecasts*.





This difference in calibration provides another practical benefit of the seamless MuTHRE model for forecast providers, in that improvements in rainfall forecasting are easily integrated into the forecasting system. Since the non-seamless monthly QPP model is calibrated using forecast rainfall, it must be recalibrated whenever a new rainfall forecast is to be used. In contrast, the seamless MuTHRE model uses only observed rainfall in calibration and does not require recalibration with different forecast rainfall, allowing for easier use of improved rainfall forecast products in operational settings.

### 2.4.3. Differences in forecasting

The differences in the model structure and calibration approach for the seamless MuTHRE model and non-seamless monthly QPP model results in key differences in terms of the forecasts that each model can produce. Figure 1(c) illustrates these differences and shows that the seamless MuTHRE model produces daily streamflow forecasts that can be used at a *range of lead times and aggregation periods*, while the non-seamless monthly QPP model produces only *one-month ahead monthly forecasts.*

## 3. Case study

### 3.1. Catchments and Data

A set of 11 catchments from the Murray Darling Basin in Australia, consisting of four catchments on the Upper Murray River (NSW and Victoria) and seven catchments on the Goulburn River (Victoria), is used in the case study. These catchments have winter dominated rainfall which leads to higher streamflow between June and October (see Figure 2), and have fewer than 5% of days with no flow. This same set of catchments was used to extensively evaluate the MuTHRE model in McInerney et al. (2020).

Time series of daily observed streamflow over a 22-year period between 1991 and 2012 are obtained from the Hydrologic Reference Stations (HRS) dataset (http://www.bom.gov.au/water/hrs). Observed rainfall and PET data over the same period are obtained from the Australia Bureau of Meteorology's climate data service (www.bom.gov.au/climate), with a climatological average used for PET (McInerney et al., 2021).

Rainfall forecasts are provided by the Australian Community Climate Earth-System Simulator - Seasonal (ACCESS-S) (Hudson et al., 2017). The post-processing method of Schepen et al. (2018) is used to reduce biases and improve the reliability of the ACCESS-S rainfall forecasts. A set of 100 replicates of post-processed forecasts that begin on the first day of each month and extend out to a maximum lead time of 1 month are used.

### 3.2. Hydrological model

The conceptual rainfall-runoff model GR4J (Perrin et al., 2003) is used as the deterministic hydrological model $h$ (introduced in Section 2.1) for simulating daily streamflow from rainfall and PET inputs. GR4J has been widely used and evaluated over diverse catchment climatologies and physical characteristics (Perrin et al., 2003; Hunter et al., 2021). GR4J represents processes of interception, infiltration and percolation, and has four calibration parameters: $x_1$ is the capacity of the production store (mm), $x_2$ is the water exchange coefficient (mm), $x_3$ is the capacity of the routing store (mm), and $x_4$ is the time parameter of the unit hydrograph (days).





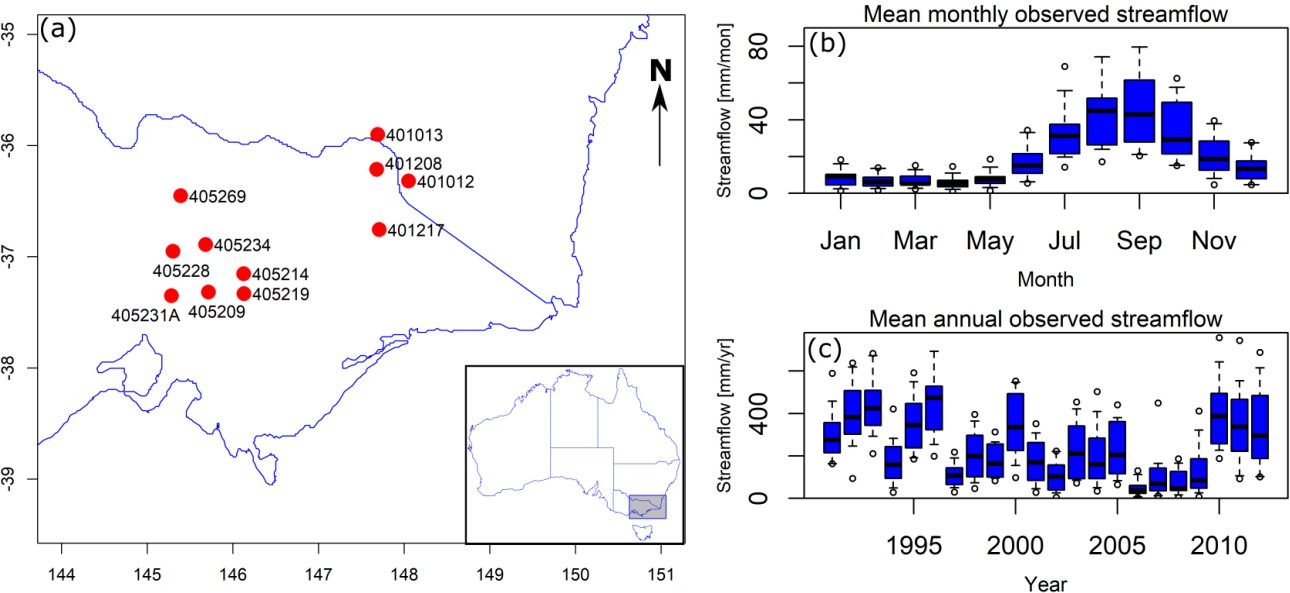

**Figure 2: Locations and mean observed streamflow for the 11 case study catchments**

### 3.3. Calibration/evaluation procedure

Calibration of model parameters and evaluation of forecasts is performed using a leave-one-year-out cross validation procedure (McInerney et al., 2020). For each year $j$, hydrological and residual error model parameters are calibrated using observed streamflow data from the entire evaluation period, except for year $j$ and the subsequent years $j+1$ to $j+4$ (which are excluded to reduce the influence of system memory on model evaluation). Hydrological model parameters are estimated using likelihood maximisation based on the BC0.2 error model (McInerney et al., 2020), implemented using a quasi-Newton optimization algorithm run with 100 independent multistarts (Kavetski and Clark, 2010). Methods for estimating residual error model parameters are described in McInerney et al. (2020) and Woldemeskel et al. (2018).

Calibrated hydrological and error models are then used to generate probabilistic streamflow forecasts for year $j$. This calibration/forecasting process is repeated for all 22 years, resulting in 22 sets of one-year forecasts, which are subsequently merged into a single 22-year forecast to facilitate evaluation against streamflow observations (as described in Section 3.4).





### 3.4. Forecast evaluation

#### 3.4.1. Performance metrics

Streamflow forecasts are evaluated using numerical metrics for the following attributes:

*Reliability,* which refers to statistical consistency between the forecast distribution and observations, is evaluated using the reliability metric of Evin et al. (2014) (which is based on the predictive quantile-quantile plot). Lower metric values are better, with 0 indicating perfect reliability, and 1 being worst reliability.

*Sharpness* refers to the spread of the forecast distribution, with sharper forecasts those with lower uncertainty. We use the

sharpness metric of McInerney et al. (2020), which is based on the ratio of the average 90% inter-quantile range (IQR) of the forecasts and a climatological distribution (described below). Lower values are better, with 0 representing a deterministic forecast (with no uncertainty) and 1 representing the same sharpness as climatology.

*Volumetric bias* refers to the long-term water balance error. It is quantified using the metric of McInerney et al. (2017) as the relative absolute difference between total observed streamflow and the total forecast streamflow (averaged over the forecast

replicates). Lower values are better, with 0 representing unbiased forecasts.

*Combined performance* is quantified using the continuous ranked probability score (CRPS) (Hersbach, 2000). We express this metric as a skill score (CRPSS) relative to the climatological distribution. Higher CRPSS values are better, with a value of 1 indicating a perfectly accurate deterministic forecast, and 0 indicating the same skill as the climatological distribution.

The *climatological distribution* represents the distribution of daily streamflow for a given time of the year based solely on

previously observed streamflow at that time of the year. The climatological distribution is constructed using a 29 day moving-window approach, described in detail in McInerney et al. (2020).

#### 3.4.2. Aggregation and stratification

The main aim of this paper is to compare the performance of the seamless MuTHRE model and the non-seamless monthly QPP model at the monthly scale. The monthly MuTHRE forecasts are obtained by aggregating daily forecasts to the monthly

scale.

*Overall* evaluation of monthly forecasts is performed using data from the entire evaluation period, i.e. all months and years, with more detailed *stratified* performance evaluation performed for individual months and years.

We also demonstrate the ability of the MuTHRE model to produce seamless forecasts, which are reliable over a range of lead times and aggregation scales. This is achieved by evaluating both (i) daily forecasts stratified by lead times from 1-28 days,

and (ii) cumulative flow forecasts for periods 1-28 days. The forecast is considered 'seamless' if reliability metrics are similar across all lead times and aggregation scales. The evaluation of cumulative flow forecasts expands on the analysis of McInerney et al. (2020), which evaluated only daily and monthly forecasts, and provides and important demonstration of seamless forecasting over the entire range of time scales between 1 and 28 days. We note that cumulative flow forecasts over 1 month correspond to monthly forecasts.





### 3.4.3. Evaluation of performance differences between QPP models

Forecast performance for the two models is compared across multiple catchments using *practical significance tests*, as described next. For each combination of performance metric (e.g., reliability, CRPSS) and stratification (e.g., month, year), a statistical test is used to determine whether differences in metric values over the range of catchments are of practical relevance. Statistical tests are performed using the paired Wilcoxon signed rank test (Bauer, 1972), with controls applied to reduce the false discovery rate (Benjamini and Hochberg, 1995; Wilks, 2006). Practically relevant differences are taken as 20% of the median metric value for the non-seamless monthly QPP model (following McInerney et al., 2020).

## 4. Results

### 4.1. Demonstration of seamless forecasting capabilities of the MuTHRE model

**Daily forecasts**

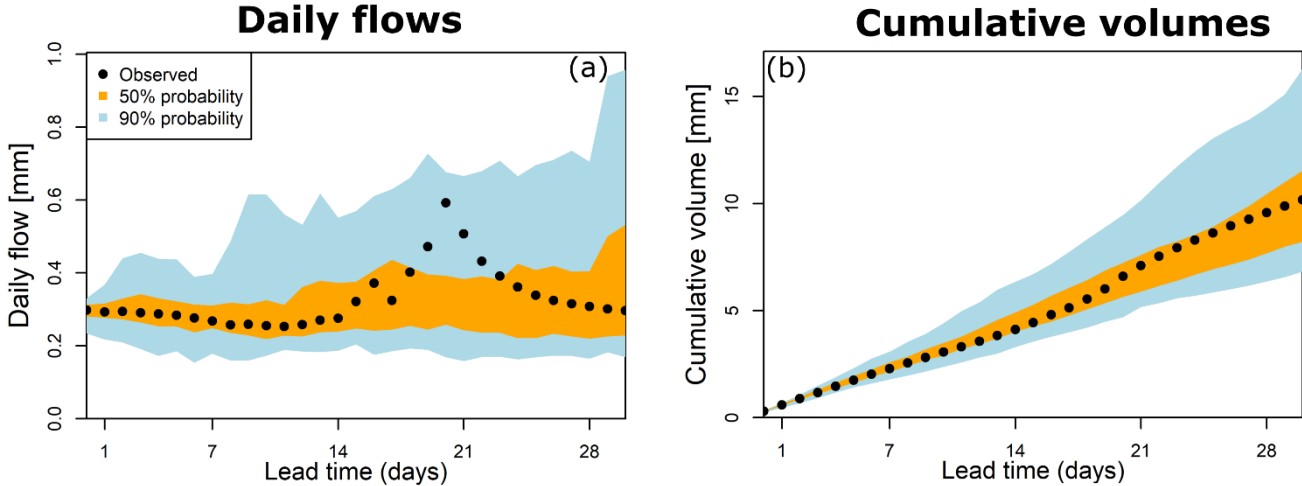

**Figure 3: Time series of daily and cumulative probabilistic forecasts from the seamless MuTHRE model for Murray River at Biggara (401012, see Figure 2) for May 2002. The non-seamless monthly QPP model does not have the capability to produce these forecasts.**

Figure 3 provides illustrations of probabilistic forecast time series in the Biggara catchment (401012, see Figure 2). Daily forecasts from the seamless MuTHRE model, beginning on 1st May 2002, are shown in Figure 3a. The observed daily streamflow lies within the 90% probability limits of the MuTHRE forecasts for each lead time. As expected, the probability limits are tight for short lead times (when forecast rainfall uncertainty and hydrological uncertainty are small), and widen for longer lead times.

Figure 4 (left column) shows performance of the daily forecasts from the MuTHRE model for lead times 1-28 days, evaluated over the full range of case study catchments. The key finding from this analysis is that reliability is relatively constant over all lead times, with median metric values lying in the tight range of 0.04-0.06 (Figure 4a). We also note that forecasts are sharper and have better CRPSS at short lead times, and that bias is relatively constant.



**Figure 4: Performance of MuTHRE forecasts in terms of daily streamflow (left) and cumulative flow (right). Metrics shown are for reliability (top row), sharpness (2nd row), volumetric bias (3rd row) and CRPSS (bottom row). Thick lines show median values and vertical bars represent 80% probability limits. Note the inverted *y*-axis for CRPSS.**





**Cumulative flow forecasts**

Figure 3b shows cumulative flow forecasts out to 28 days in the Biggara catchment for the representative time period. The cumulative flows based on observed streamflow lie well within the 90% probability limits of the MuTHRE forecasts for all lead times.

Figure 4 (right column) shows performance of the cumulative flow forecasts from the MuTHRE model for lead times 1-28 days over all catchments. Again, we see that reliability is relatively constant over all lead times, with median metric values 350 between 0.04 and 0.06 (Figure 4b). We also note that sharpness, volumetric and CRPSS metrics are typically better for cumulative forecasts than daily forecasts (compare left and right columns in Figure 4).

In summary, the forecasts from the MuTHRE model are seamless, since they are reliable over (a) the range of lead times, and (b) multiple aggregation scales, from the shortest scale of 1 day, to the longest of 1 month, and everything in between. This confirms and extends previous findings in McInerney et al. (2020). In contrast to the seamless MuTHRE model, the non- 355 seamless monthly QPP model does not have the capability to produce forecasts of daily streamflow and cumulative flows for time periods less than one month.

### 4.2. Comparison between monthly forecasts

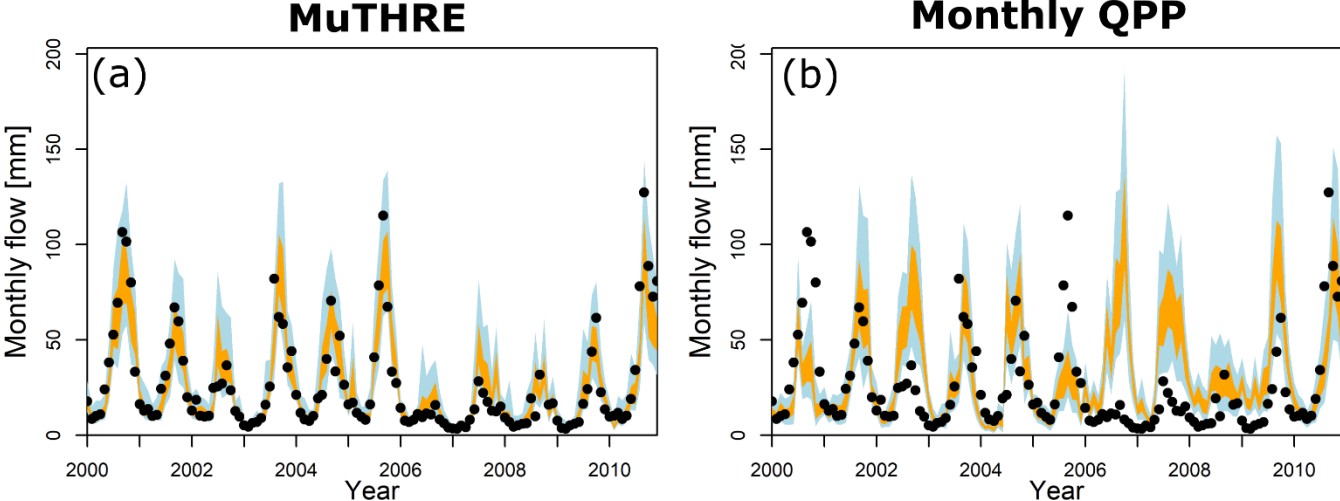

**Figure 5: Time series of monthly probabilistic forecasts for Murray River at Biggara (401012, see Figure 2) for the seamless**
**MuTHRE model and non-seamless monthly QPP model. Results are shown between the years 2000 and 2011.**

Figure 5 compares monthly forecasts from the seamless MuTHRE model and non-seamless monthly QPP model for the Biggara catchment. For high flow periods the 90% prediction limits from the seamless MuTHRE model appear to be tighter (i.e. have less uncertainty) than those from the non-seamless monthly QPP model, while still capturing the observed streamflow.



The monthly forecasts from the MuTHRE and monthly QPP models are compared in Figure 6 in terms of overall performance (left column), and when stratified by month (middle column), and year (right column). The key findings are as follows.

**Reliability**. Figure 6a shows that the overall reliability of monthly forecasts from the MuTHRE and monthly QPP models is similar. While the median metric value is 0.06 for the seamless MuTHRE model is larger than the median value of 0.04 for the non-seamless monthly QPP model, these differences are not practically significant. Figure 6b shows that when performance is stratified by month, the two models have similar reliability (i.e. not practically significant) for all 12 months. When stratified by year, the MuTHRE model offers similar reliability to the monthly QPP model for 20 out of the 22 years, with the non-seamless monthly QPP model offering practically significant improvements in 2 of the 22 years (Figure 6c).

**Sharpness.** Figure 6d shows that the overall sharpness of monthly forecasts from the seamless MuTHRE model is slightly better than the non-seamless monthly QPP model (median metric values of 0.44 c.f. 0.49), although differences are not practically significant. Figure 6e shows that when sharpness is stratified by month, the seamless MuTHRE model provides practically significant improvement in 1 month (September) and similar performance in the other 11 months. Figure 6f shows sharpness stratified by year is similar for both models for all years.

**Volumetric bias**. Figure 6g shows that the overall volumetric bias from both models is similar (median of 0.01). Figure 6h shows that when stratified by month, the MuTHRE model produces similar/better performance in all months, with practically significant improvements in December and similar performance in the remaining 11 months. Figure 6i shows that when stratified by year, the MuTHRE model produces similar/better performance in 19 out of 22 years, with practical significant improvements in one year, while the monthly QPP model provides practically significant improvements in 3 years.

**CRPSS**. In terms of overall CRPSS, Figure 6j shows that the seamless MuTHRE model (median metric value of 0.45) provides slight improvement over the non-seamless monthly QPP model (median metric value of 0.42), although these differences are not practically significant. Figure 6k shows that when stratified by month, the seamless MuTHRE model actually provides similar performance in all 12 months. Figure 6i shows that when performance is stratified by year, the seamless MuTHRE model actually provides practically significant improvements in CRPSS in 2 out of 22, and similar performance in the remaining 20 years.

In summary, aggregated forecasts from the seamless MuTHRE model offer similar (not practically significant), and in some cases superior performance, to forecasts from the non-seamless monthly QPP model, for the vast majority of performance metrics and stratifications considered in this study.



**Figure 6: Monthly performance of the seamless MuTHRE and non-seamless monthly QPP forecasts for all data (left), stratified by month (middle) and stratified by year (right). Results are shown for reliability (top row), sharpness (2nd row), volumetric bias (3rd row) and CRPSS (bottom row). Circles/squares indicate that the MuTHRE model performs practically significantly better/worse than the monthly QPP model**





## 5. Discussion

### 5.1. Interpretation of key findings

The empirical results show that the seamless MuTHRE model achieves essentially the same performance as the non-seamless monthly QPP model at the monthly time scale, and even provides improvement in some aspects. At first glance, this outcome may seem surprising for the following reasons:

- The seamless MuTHRE model is required to produce reliable forecasts over a range of lead times and aggregations scales, whereas the non-seamless monthly QPP model is only required to produce monthly streamflow forecasts.

- The seamless MuTHRE model is calibrated at the daily scale, using only observed daily streamflow during calibration, while the non-seamless monthly QPP model is calibrated to match the observed monthly streamflow.

- The seamless MuTHRE model does not see the forecast rainfall during calibration, whereas the non-seamless monthly QPP model does.

The subsections below describe how the seamless MuTHRE model is able to achieve comparable/better performance than the

non-seamless monthly QPP model despite these apparent challenges.

#### 5.1.1. Time scale of forecasting/calibration

The seamless MuTHRE model produces daily forecasts that can be aggregated from time scales of one day to one month, whereas the non-seamless monthly QPP model produces forecasts only at the monthly scale. One might expect the enhanced capability obtained from the seamless MuTHRE model to come at some cost in performance at the monthly scale.

Encouragingly, this is not the case.

The ability to reliably aggregate daily forecasts to the monthly scale demonstrates that the seamless MuTHRE model is adequately capturing temporal persistence in daily forecasts. The MuTHRE model represents temporal persistence in *hydrological errors* using the daily AR(1) model, and the (30-day) dynamic bias component. This is important because neglecting temporal persistence in *hydrological errors* can result in an underestimation of hydrological uncertainty for

aggregated predictions/forecasts (Evin et al., 2014). The reliability of aggregated forecasts also suggests that the (post-processed) *rainfall forecasts* are capturing the temporal persistence of observed rainfall required to produce reliable monthly rainfall forecasts (see Section 5.1.2).

The seamless MuTHRE model is not calibrated to optimize performance at the monthly scale, since it uses only observed daily streamflow during calibration. On the other hand, the non-seamless monthly QPP model is calibrated to match the observed

monthly streamflow, which could lead to improved performance at the monthly scale compared with the seamless MuTHRE model. The performance at the monthly scale is particular impressive since monthly data is not used in calibration.

#### 5.1.2. Use of observed vs forecast rainfall used in calibration

The residual error model in the non-seamless monthly QPP model represents combined rainfall and hydrological uncertainty. It uses forecast rainfall during calibration, and can (in theory) correct for biases and under/over-dispersion in rainfall forecasts.

In contrast, the seamless MuTHRE model represents only hydrological uncertainty, and is calibrated using observed rainfall. Uncertainty due to forecast rainfall is represented by propagating rainfall forecasts through the hydrological model. Since the





MuTHRE model does not correct for forecast rainfall errors, this approach relies on rainfall forecasts being reliable in order to produce reliable streamflow forecasts (Verkade et al., 2017).

In this study we have used rainfall forecasts from the ACCESS-S climate simulator (Hudson et al., 2017), which were post-
processed at the catchment scale with the aim of reducing biases and over/under-dispersion at the daily scale, and capturing the temporal persistence in rainfall (Schepen et al., 2018). As a result, these post-processed rainfall forecasts are reliable and sharp at both the daily and monthly scale, and do not have a detrimental impact on the performance of the seamless MuTHRE model.

### 5.1.3.    Use of daily vs monthly streamflow observations to update forecasts

The MuTHRE and monthly QPP models differ in their use of recently observed streamflow data. The MuTHRE model uses daily streamflow observations to update both (i) the dynamic bias component of the error model, to account for monthly errors, and (ii) the daily AR1 model to account for recent daily errors and improve sharpness of forecasts for short lead times. In contrast, the monthly model only uses monthly aggregated streamflow observations to update the monthly AR1 model. The ability to utilize the most recent time series of daily streamflow observations provides the MuTHRE model with a potential
advantage over the monthly QPP model (which see only monthly totals) and may be another reason why the MuTHRE model performs so well compared with the monthly QPP model.

### 5.1.4.    Summary of practical benefits of seamless forecasts

The key practical benefits of these differences are summarized below:

1. Seamless forecasts can be used to inform decisions at a range of time scales
2. Seamless daily forecasts are easily integrated into river system models used for real-time decision-making
3. Simplifies forecasting system as a single seamless product can serve range of forecast requirements at different time scales.
4. Improvements in rainfall forecasting are easily integrated into the forecasting system.

The competitive performance of the MuTHRE model even at the native scale of the non-seamless monthly forecasts is clearly encouraging and, in combination with the inherent benefits listed above, provides further motivation to adopt seamless
forecasts in research and practical work.

### 5.2.  Future work

Future work in this area will focus on the following aspects:

- Further testing and development of the MuTHRE model on a wide range of catchments. The monthly QPP model has been comprehensively evaluated on 300 catchments around Australia (Woldemeskel et al., 2018), whereas the MuTHRE model
has currently been evaluated on 11 catchments in the Murray Darling Basin. Evaluation of the MuTHRE model over a wide range of hydro-climatic conditions is required to ensure the findings of this study are robust. Potential modifications to the MuTHRE model, including the treatment of zero flows in ephemeral catchments (McInerney et al., 2019; Wang et al., 2020), may be required to ensure the MuTHRE model remains competitive with the monthly QPP model over a wider range of catchment types.



• Deeper understanding of the specific reasons for MuTHRE model matching the monthly QPP model at the monthly scale. Further work could look at different combinations of MuTHRE/monthly QPP model components to systematically diagnose the reasons why the MuTHRE model performs so well. This was not feasible in this study.

• Evaluating how the use of different rainfall forecasts, with differing quality, impacts on the performance of the seamless MuTHRE model and its ability to match/improve on the performance of the non-seamless monthly QPP model at the

monthly scale.

## 6. Conclusions

This study has explored the question of whether aggregated seamless forecasts from the daily Multi-Temporal Hydrological Residual Error (MuTHRE) model can produce similar performance at the monthly time scale to forecasts from the non-seamless monthly streamflow post-processing (QPP) model used in the Australian Bureau of Meteorology's Dynamic

Forecasting System. A case study with 11 catchments in south-east Australia, the GR4J conceptual rainfall-runoff model, and post-processed ACCESS-S rainfall forecasts, is reported.

The key finding is that the seamless MuTHRE model is able to achieve essentially the same performance as the non-seamless monthly QPP model for the majority of metrics (reliability, sharpness, bias and CRPSS) and stratifications (monthly and yearly). Remarkably, the seamless model achieves high quality forecasts at its native daily scale and matches the performance

of the non-seamless monthly model at the monthly scale (despite not being calibrated at that time scale).

Seamless subseasonal forecasts, which are reliable over a wide range of lead times (1-30 days) and time scales (daily-monthly), offer numerous practical benefits over non-seamless forecasts. For users, seamless subseasonal forecasts can inform a wide range of management decisions from flood warning to water supply operation, while for service providers, seamless forecasts will reduce the number of forecast products that require development and operation. As such it represents a single modelling

tool with great versatility. The clearly encouraging results from this study help pave the way for seamless forecasts to replace non-seamless forecasts, as they offer additional capability without loss in performance at the time scale of non-seamless forecasts.

## Acknowledgments

The recent presented in this paper was funded by the Australian Bureau of Meteorology. Support with supercomputing

resources was provided by the Phoenix HPC service at the University of Adelaide. Data used in the case studies is available as follows: observed rainfall data from www.bom.gov.au/climate, observed streamflow data from www.bom.gov.au/waterdata, and post-processed ACCESS-S forecast rainfall from https://doi.org/10.25909/14604180. We gratefully acknowledge insightful feedback from Surendra Rauniyar and Christopher Pickett-Heaps during the Bureau review of this manuscript.



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
