# Peer review of "Seamless streamflow forecasting at daily to monthly scales: MuTHRE lets you have your cake and eat it too"

_Hydrology and Earth System Sciences, 2021_

## Author Comment (AC1)

**Response to comments from Reviewer 1**

1.1 I found this manuscript to be of high quality, interesting and very well written. In particular, I find the figures very clear and of high quality. I have a few minor comments that I would like to see addressed, but this should not require a lot of time.

We thank Reviewer 1 for their encouraging feedback, and for the valuable suggestions that will improve our paper.

1.2 When I first started reading the manuscript, including the title, I was very confused by what you refer to when you talk about a "model". To me, and I think for most hydrologists, our first reflex is to think about the hydrological model (GR4J in your case). It later became clear that you were most of the time referring to the post-processing method, but I was still bothered throughout the paper by the fact that it could be clearer. For one thing, I would suggest changing the title to indicate that you are interested by comparing two post-processing methods.

We agree that our use of "model", when referring to "post-processing model", was too vague. In particular, the phrase "streamflow model" in the title of the paper was a poor choice. We apologise for the confusion this caused.

There are several models used in the work, including the deterministic model, the residual model, etc. The usage of "model" in the original manuscript was a bit loose and we can see how this caused confusion.

For the revised paper we will
i. Clarify specifically which models we refer to in specific sentences, to avoid ambiguity/ confusion
ii. Change the title to clearly indicate the study focuses on streamflow forecasting and compares two post-processing methods

1.3 Second, I would avoid the use of the QPP acronym and instead refer to the "post-processing method" or "post-processing model". I recognize that it would be a bit longer, but in my opinion it would be clearer. I'm one of those persons who don't like acronyms too much...

We agree that the use of acronyms can often make it difficult to follow the text. On the other hand it can also lead to more compact wording. We will review the usage of acronyms across the paper and reduce them where possible.

1.4 Section 2 should be named "Post-processing methods" or "Post-processing models", and maybe the titles of the subsections could be adjusted accordingly.

We do agree that the loose usage of word "model" across the paper created considerable confusion.

The current title "Forecasting Models" is intended to make it clear we are talking about forecasting methods. A title "Post-processing" on its own does not communicate this.

In addition, the section content is not limited to post-processing methods – it also includes the deterministic hydrological model itself. It would be strange and potentially confusing in its own right to introduce the deterministic model in a section titled "post-processing" methods.

We do need some clear term for the complete system, which includes the deterministic model plus the post-processing model. We currently believe "Forecasting model" is an appropriate term, once it is clearly defined in the revised manuscript. However this will be reviewed as part of the comprehensive tightening of terminology across the paper

1.5    Finally, I would suggest mentioning GR4J explicitly in section 2.1 and referring to section 3.2 for greater details. In fact, I was initially a bit bothered by the fact that GR4J was described in the case study and not in the "Forecasting models" section, because to me, GR4J is a model and it can be used jointly with meteorological forecasts to produce streamflow forecasts. I think I understand what you wanted to do here: you wanted to avoid drawing too much attention to GR4J because it is not the central element of your study. So in the end, I don't mind too much GR4J being described in section 3.2, but I really think it would help clarify/distinguish things if you could mention it in section 2.1 too.

We do not focus on GR4J in Section 2.1 because the streamflow post-processing models can be used with any hydrological model, not just GR4J. As such, we simply referred to a "deterministic hydrological model $h$" in Section 2.1 and introduce GR4J in the case study in Section 3.2.

However, we do see value in providing some context regarding the term "deterministic hydrological model", and will refer to GR4J in Section 2.1 as an example of such deterministic model used in this study. We will revise the text accordingly.

1.6    This is linked to point 1. When I read the abstract and intro, I had a bit of difficulty understanding what you meant by "designed" for a single time step", because I had, again, a hydrological model in mind. So I was think that, for instance, if GR4J runs at a daily time step, there is nothing stopping you from aggregating it at the monthly time step. So I wasn't completely sure were you were going with that, until I read section 2. It then became very clear that the monthly post-processing method is completely inapplicable at the daily time step and I understood what you meant in the abstract and intro. I think it would be better it could be clearer right from the start, and I think that making it more explicit that you refer to post-processing methods would really help.

Thanks for highlighting this problem. We see that this would have been confusing.

In the revised paper we will explicitly refer to "post-processing models" when we state that forecasts are commonly designed for a single time step (i.e. in the first paragraphs of the abstract and the introduction).

See also response to comment 1.2 on the use of term "model" across the paper.

1.7    Page 1 line 20: Would it be more accurate to refer to ACCESS-S as an atmospheric model? When I think of a climate model, I think about the ones used for climate change study, that really model the climate over long periods and for which the fluctuations at smaller time steps have no real significance. I know they are basically very similar in their structures, but if I understand correctly, they are not run in the same way at all

The reviewer is correct that "climate model" is not the appropriate term. We will refer to ACCESS-S as a "numerical weather prediction model" in the revised paper.

1.8    I was really thankful for Figure 1. Section 2 was a bit heavy, but this figure is really helpful to understand how everything is put together. Great work.

Thank you.

1.9 Very small point: is it possible that the acronym REM is not defined in the text? Maybe I missed it? I understand it means Residual Error Model but can you please make sure it is defined in the text when first mentioned? Just for clarity.

Good catch. We will make sure this term is defined.

1.10 Very very small point: Line 111, why chose the subscript "foc"? Why not "fcst" if it is forecasted? I think it would be clearer?

"foc" is an abbreviation of "forecast" and is shorter than "fcst". We will review the usage of acronyms across the paper.

1.11 Page 5 line 141: do you mean each individual member or is it really each forecast? I understand you reduce (or "collapse") the forecasts to deterministic ones, so I guess it is really "each forecast", but I'm not sure. Can you maybe clarify? I was a bit lost.

We should have stated that the error model is applied to each individual member (i.e. each ensemble member). We will clarify this in the text.

Note for the MuTHRE model we do not collapse the ensemble of forecasts to a deterministic forecast. We see why the previous wording may have suggested otherwise, and believe the revised text will overcome this confusion.

1.12 Page 6 line 167: I like that you use the word "collapse" when referring to the transition from ensemble to deterministic. I think I will adopt this terminology myself in the future.

Nice.

1.13 Section 2.4 is very helpful. Very clear. Thank you.

Thank you!

1.14 Page 10 line 258: I would suggest referring to Schepen et al (2018) as a pre-processing method instead of a post-processing method, especially in the context of your study, since you use the result from this method as an input.

Good point. We will now say that "the rainfall forecasts are pre-processed using the method of Schepen et al (2018)" and use similar wording elsewhere in the paper.

1.15 I like section 3.4.1. The choice of performance indicators is very well justified and they are clearly appropriate.

Thanks.

1.16 I was surprised that Figure 5 b was not discussed more, especially in relations to the findings in the next page (statistical significance for the different metrics. When looking at Figure 5b, I can't help but think that the monthly post-processing method is not doing a very good job, and this is a bit surprising given that it is trained for monthly forecasts. First, there is an underestimation of streamflow between 2000-2001, and then (2006-2008) a pretty large overestimation. Why is that? Any idea? Then later we learn that, according to the different performance metrics, the differences in performance are actually not often statistically significant. I was surprised by that, because of that figure. I would really appreciate a bit more discussion/explanation of what is happening in 2000-2001 and 2006-2008.

Reviewer 1 is correct that the time series presented in Figure 5b for the monthly QPP model are surprising, and do not appear consistent with the metrics in Figure 6.

It turns out that there was a bug in the code used to produce this time series plots for the monthly QPP model.

The correct time series in Figure 1 below (which we will use in the revised paper) shows very little difference between monthly forecasts from the MuTHRE and monthly QPP models. This is much more consistent with metrics presented in Figure 6 of the paper.

[Figure]

*Figure 1: **Revised Figure 5** for the manuscript, showing time series of monthly probabilistic forecasts for Murray River at Biggara.*

We are very thankful to Reviewer 1 for their attention to detail, which has allowed us to identify and resolve this problem.

1.17 Line 388: I suggest removing the word "actually" to avoid repetition with the previous sentence

Will do.

---

## Author Comment (AC2)

**Response to comments from Reviewer 2**

**Summary**

2.1   In this paper, the authors introduce a Multi-Temporal Hydrological Residual Error (MuTHRE) model that enables the production of seamless streamflow forecasts (e.g., daily, weekly, fortnightly, monthly) within the range of 1-30 days. The approach is described and compared against a non-seamless streamflow post-processing (QPP) model implemented by the Australian Bureau of Meteorology's Dynamic Forecasting System. The comparison is performed in 11 Australian catchments in terms of several forecast attributes, and the authors conclude that the MuTHRE model is not only capable of providing good performance for daily streamflow forecasts and cumulative volumes, but also similar performance to that obtained with the non-seamless QPP model for monthly flows.

Overall, this is an interesting manuscript that contributes with encouraging results on the use of seamless streamflow forecasting frameworks. The motivation is clearly stated and the results are nicely presented. There is, nevertheless, a lot of room for improving explanations of the model formulation, streamflow forecast generation, and verification, so that any reader could reproduce the results presented here. There are other minor comments and editorial suggestions that may also help the authors to improve the quality of their manuscript.

We thank Reviewer 2 for their encouraging feedback and detailed review of our paper. In particular, we appreciate their thoughtful suggestions for improving the description of streamflow post-processing models in Section 2, which will make this material easier to follow.

**Main comment:**

2.2   Model description (section 2): I found this section very hard to understand. I think the manuscript would greatly benefit from re-organizing the material and improving definitions and descriptions. For example:

It seems that the two approaches compared here follow the same general model structure (equations 1 and 2). Is that what you mean with "both QPP models"? Can you please be more explicit?

We thank Reviewer 2 for their encouraging feedback and detailed review of our paper. In particular, we appreciate their thoughtful suggestions for improving the description of Yes the two approaches have the same general structure. We will rewrite this sentence to be more explicit about this.

We will also perform a comprehensive review of terminology across the paper, which will also help avoid confusion in this section.

2.3   Also, Qt is described as a "probability model for streamflow" (equation 1), and then as a "residual error model" (L103, equation 2) when it is, in reality, the sum of deterministic model output and a residual error term. I wonder if you actually need equation (1) in this description.

The sentence describing $Q_t$ as a residual error model was indeed incorrect and confusing - thanks for highlighting this error. As the reviewer notes, $Q_t$ is the sum of deterministic model output and a residual error term (which is modelled using a residual error model). We will rectify this in the revised paper.

Equation 1 is important in summarising the overall structure of the forecasting models, notably their probabilistic nature, their inputs and parameters.

2.4 I think it would be better to have the information presented in L191-214 (differences between MuTHRE and monthly QPP model) right after section 2.1.

We appreciate this suggestion.

A major benefit would be that Figure 1a, which shows structures of two post-processing models, is presented earlier on. This would make it easier to follow the descriptions of the post-processing models in Sections 2.2 and 2.3.

However, moving all of the information in Lines 191-214 (Section 2.4.1) to immediately after Section 2.1 would introduce a new set of problems, because the text in Section 2.4.1 relies on material presented in the earlier Sections 2.2 and 2.3.

Based on these considerations, we will
   i.   Introduce Figure 1a much earlier on, when we first describe the overall approach for the MuTHRE and monthly QPP models (Sections 2.2.1 and 2.3.1).
   ii.  Keep most of the summary of structural differences between the post-processing models in Section 2.4.1, in order to avoid forward referencing.

2.5 The authors should consider separating Figure 1 (which is very nice) into two figures: one for model structure (which could include model equations for more direct comparisons between model structures), and another figure for model calibration and forecasting.

There are pros and cons to this suggestion.

Separating out the model structure figure would allow us to present this schematic earlier, and make it easier to follow the description of the post-processing models (see comment 2.4 above). On the other hand the schematics can be more easily compared to each other when they are presented as multiple panels within the same figure.

In terms of including equations in the figure, it is not immediately clear whether it is feasible to include all relevant equations in this figure without creating clutter and potential confusion. Note that the key differences between the post-processing models – namely aggregating to monthly scale and taking the median – are better described schematically rather than through equations.

2.6 The meaning of z should be included after presenting equation (2) (perhaps in line 107).

Good suggestion. We will move the meaning of $z$ earlier, so that it comes directly after equation (2).

2.7 Since Xt is also used to describe state variables in the hydrology literature (especially in data assimilation books/papers), I think ut would be more appropriate for meteorological forcings (e.g., Liu and Gupta 2007).

Thanks for this suggestion. Ideally our notation would be consistent with the broad hydrological literature. However, since the previous papers on the MuTHRE model (McInerney et al., 2020;

McInerney et al., 2021) use $\mathbf{x}_t$ for meteorological forcings we prefer to also use this notation for consistency.

2.8 Additionally, in L125 you describe st as a time-varying scaling factor, while the same variable is used to describe hydrological model states in L100.

Good catch. We will change the symbol for the time-varying scaling factor

2.9 L113: I presume that the raw streamflow forecasts do not account for uncertainty in hydrologic model parameters. Can you please clarify?

Correct. We will clarify this.

2.10 L135: Is the ensemble size still Nfoc after adding the residual term?

Yes it is. We will clarify this.

2.11 L141: What do you mean by "individual raw forecast"? Each ensemble member produced with the ensemble of rainfall forecasts?

Yes this is what we meant. We will ensure that consistent terminology is used for ensemble members throughout the paper.

2.12 L148: how is m* determined?

It is computed as the mean of the residual $\boldsymbol{\eta}$ after the seasonality and dynamic bias terms are removed. We will mention this in the text.

2.13 Since the paper should be self-contained, additional information on the calibration procedures referred to in L162 and L190 should be provided (what are the calibration period, objective functions, and optimization algorithms?). A couple of sentences should suffice.

We agree with the reviewer that additional information on the approaches used to calibrate the post-processing models should be included here. In the revised text we will describe how the parameters in the streamflow post-processing models are estimated (e.g. method of moments, maximum likelihood).

We note that Section 3.3 provides details of cross-validation procedure (including calibration periods), as well as the objective function and optimization algorithm for calibrating the hydrological model.

2.14 L167: "and then collapsed to a deterministic forecast by taking the median". Is this current operational practice?

This step is performed in the Bureau of Meteorology's dynamic forecasting system (see Section 2.3.2 in Woldemeskel et al., 2018).

2.15 L111, L112, L135, L168, L169, and elsewhere: is "replicate" the same as "ensemble member"?

Yes "replicate" is the same as "ensemble member". We will ensure "ensemble member" is used throughout paper

**Additional minor comments**

2.16 L33-35: It makes more sense to me to describe common practice before referring to the need for seamless forecasts. Also, it would be worth highlighting that non-seamless forecasting efforts have been (and are being) conducted in South America (e.g., Souza Filho and Lall 2003; Mendoza et al. 2014), Europe (e.g., Ionita et al. 2008; Hidalgo-Muñoz et al. 2015), Asia (e.g., Pal et al. 2013) and everywhere else around the world, with appropriate citations.

Thanks for this suggestion. We originally referred to "seamless forecasts" first, because it made it easier to define a "non-seamless forecast". However, it does make sense to start with common practice, and we plan to implement this change.

We also appreciate the references for other non-seamless forecasts around the world, and will include these in the introduction where appropriate.

2.17 L37: "This is the focus of our study". This reads out of place here. I recommend deleting this sentence or moving it toward the end of the introduction.

We will remove this sentence.

2.18 Figure 2: How many values are contained in each boxplot? One per basin?

There are 11 values in each boxplot, corresponding to the 11 catchments. We will clarify this in the caption.

2.19 Since you have only 11 catchments, I think it would be better to show one line per basin.

Thanks for the suggestion. We did try this, but preferred the boxplots because
   - They are cleaner (fewer lines), which makes it easier to see variability between months and years
   - All catchments show roughly the same patterns for both monthly and annual variability, so there is not much to be gained from showing each catchment

2.20 Further, it would be informative for readers to have a table with the name of the station, basin-averaged elevation, area, mean annual runoff, mean annual precipitation, mean annual temperature, annual runoff ratio, and aridity index.

Good suggestion. We intend to include a table will relevant catchment information in the revised manuscript.

2.21 L273: Are you working with calendar years or water years?

Calendar years. We will clarify this in the text.

2.22 Are daily forecasts produced each day in year j with MuTHRE, or only at the beginning of each month?

Forecasts for the MuTHRE model are produced at the start of the month only. We will clarify this in the text.

2.23 What is the final ensemble size of your forecasts?

The size of the post-processed streamflow ensembles is 100, which is the same as size of the forecast rainfall ensembles. We will clarify this in the text.

2.24 L274-275: The problem of hydrologic memory in Australian catchments and its implications for cross-validation has been previously documented (e.g., Robertson et al. 2013; Pokhrel et al. 2013). I recommend the authors read and cite these papers here. The following blog article is also relevant: https://hepex.inrae.fr/how-good-is-my-forecasting-method-some-thoughts-on-forecast-evaluation-using-cross-validation-based-on-australian-experiences/

Thank you for these references. We will add appropriate citations regarding the importance of dealing with hydrologic memory in the cross-validation procedure.

2.25 L292: Perhaps it would be better to replace the word "uncertainty" with "spread".

Good suggestion. We will make this change.

2.26 Also, it would be informative to state that sharpness is a forecast attribute only (i.e., it does not depend on the observations).

Good suggestion. We will add this.

2.27 L296: Since CRPS measures the difference between forecast and observation CDFs, it would be better to refer to "probability forecast errors" instead of "combined performance".

We describe CRPS as a metric for "combined" performance because, as shown in Hersbach (2000), the CRPS can be decomposed into terms representing individual performance aspects, namely reliability, and uncertainty/resolution (related to sharpness). We agree this may not be obvious to a general reader and will clarify this in the text. We will also clarify that for a single observation, the CRPS represents the error between the forecast distribution and the observed data point.

Note that the CRPS serving as a combined performance metric is very relevant to our study because the other three metrics – namely reliability, sharpness and bias - focus on fundamentally more specific performance characteristics. If we do not highlight this, readers may form the erroneous impression that we have four independent performance metrics.

2.28 Figure 4: How are confidence limits generated?

These are $10^{th}$ and $90^{th}$ percentiles of metric values based on values for the 11 catchments. We will clarify this in the caption.

2.29 Do you compute the metric merging forecasts from all basins? Please clarify these points in the manuscript.

The metrics are computed separately for each catchment, and the distribution of these metric values is summarized using the median, $10^{th}$ and $90^{th}$ percentiles. We will clarify this in the caption.

2.30 L368-370: You mention that reliability results are similar, although the boxplots look different. I recommend applying a statistical test to determine whether the distributions of these metrics are significantly different.

We do use a statistical test to evaluate difference in distributions of metric values. Specifically, we use "practical significance testing" (described in Section 3.4.3) to determine whether differences between models are not just statistically significant, but are statistically significantly larger than some pre-defined practically relevant margin (chosen as 20% of metric value).

Although the distributions in Figure 6a appear different, these differences are not practically significant based on the criteria defined in Section 3.4.3.

2.31 L370 and elsewhere: "practically significant" or "significant". Are the authors referring to a statistically significant result? If not, I suggest re-wording or deleting the word 'significant'.

Following on from the above point (2.30), the term "practically significant" refers to cases where the difference in metric values are statistically significantly larger than a pre-defined margin.

We note that we did not explicitly define the term "practically significant differences" in Section 3.4.3, which may have led to confusion.

We will make the following changes in order to make it clearer what "practically significant" refers to
1. We will explicitly define "practically significant" in Section 3.4.3
2. We will refer to Section 3.4.3 when we first mention "practically significant" in the Results section.

2.32 Figure 6: I think you should say "overall monthly performance" in the caption, and perhaps remind readers here what "overall" means.

This is a good suggestion. We will do this.

2.33 Are you grouping the results of all basins? In the left panels, how many points are contained in each boxplot?

Yes we are showing the results from all catchments in the boxplots. Each boxplot represents the distribution of metric values from the 11 catchments. We will clarify this in the caption.

2.34 In the center and right panels, how are the confidence limits computed?

These are $10^{th}$ and $90^{th}$ percentiles based on values for the 11 catchments. We will clarify this in the caption.

2.35 L421: Shall we expect persistence in rainfall, given the chaotic nature of the atmosphere?

Good question.

In this sentence we are
 - Referring to the day-to-day persistence in rainfall, which will be important for reliable monthly rainfall forecasts.
 - Not referring to longer-term persistence, which will be much smaller due to the chaotic nature of the atmosphere.

We will clarify that we are referring to day-to-day persistence in the revised paper.

2.36 L426: I encourage the authors to replace the last sentence of this paragraph (which reads a lot like "propaganda") with a more quantitative statement regarding the performance of MuTHRE.

We will rewrite this sentence.

2.37 L467: "This was not feasible in this study". If you cannot provide an explanation on why was not feasible, I suggest deleting this sentence.

We will remove this sentence.

2.38 L479: "High-quality forecasts". Note that the quality depends a lot on the forecast attributes you are analyzing. I think it would be good to provide a brief discussion (maybe in section 5) about tradeoffs between the metrics included here (e.g., how your forecast system can improve reliability at the cost of losing sharpness), and what makes a forecast "good" or "high-quality".

We agree that our classification of "high-quality forecasts" is dependent on the forecast attributes considered in this study. We will make this point by changing this phrase to "high-quality forecasts, based on the metrics considered in this study" (or something similar).

The results of our case study show that differences in reliability for the two models are not practically significant. As such, there is no (practically relevant) trade-off between the reliability and sharpness of forecasts to discuss.

**Suggested edits**

2.39 L51: 'Hydro-electric' -> 'hydropower'.
L70: 'drop in' -> 'loss of'.
L312: 'which' -> 'who'.
L377: 'in 1 month (September)' -> 'in September'.
L380: delete 'similar/better performance in all months, with practically'.
L382: delete 'similar/better performance in 19 out of 22 years, with practical'.
L451: 'Simplifies' -> 'A simplified'.
L473: 'to forecasts' -> 'compared to forecasts'.

Thank you for these suggested edits. We intend to implement these changes.

**References**

Hersbach, H. 2000. Decomposition of the Continuous Ranked Probability Score for Ensemble Prediction Systems. *Weather and Forecasting,* 15**,** 559-570.
McInerney, D., Thyer, M., Kavetski, D., Laugesen, R., Tuteja, N. & Kuczera, G. 2020. Multi-temporal hydrological residual error modelling for seamless sub-seasonal streamflow forecasting. *Water Resources Research,* 56**,** e2019WR026979.
McInerney, D., Thyer, M., Kavetski, D., Laugesen, R., Woldemeskel, F., Tuteja, N. & Kuczera, G. 2021. Improving the Reliability of Sub-Seasonal Forecasts of High and Low Flows by Using a Flow-Dependent Nonparametric Model. *Water Resources Research,* 57**,** e2020WR029317.
Woldemeskel, F., McInerney, D., Lerat, J., Thyer, M., Kavetski, D., Shin, D., Tuteja, N. & Kuczera, G. 2018. Evaluating residual error approaches for post-processing monthly and seasonal streamflow forecasts. *Hydrol. Earth Syst. Sci. Discuss.,* 2018**,** 1-40.

---

## Author Response (AR1)

Dear Associate Professor Werner,

We thank you for handling the editorial process for our manuscript, and the two reviewers for their constructive comments and suggestions.

We have taken on board the vast majority of the reviewers' suggestions, and have revised the manuscript accordingly. The most notable improvements to our manuscript have come from:

1. Rewriting and reorganising Section 2 (Forecasting Methods) based on comments from Reviewer 2 (see detailed response to comments 2.2-2.17 below)
2. Removing ambiguity about the term "model", based on comments from Reviewer 1. This includes changing the title of the paper to avoid the confusing term "streamflow model" (comment 1.2).

In our detailed response, we respond to individual comments from the two reviewers. We have itemized all the comments for ease of reference, and our responses are in red text for your reading convenience.

We once again thank the reviewers for their many helpful comments and suggestions which have substantially improved the quality of our manuscript.

Yours sincerely,

David McInerney and co-authors

12 July, 2022

**Response to comments from Reviewer 1**

1.1 I found this manuscript to be of high quality, interesting and very well written. In particular, I find the figures very clear and of high quality. I have a few minor comments that I would like to see addressed, but this should not require a lot of time.

We thank Reviewer 1 for their encouraging feedback, and for the valuable suggestions that have improved our paper.

1.2 When I first started reading the manuscript, including the title, I was very confused by what you refer to when you talk about a "model". To me, and I think for most hydrologists, our first reflex is to think about the hydrological model (GR4J in your case). It later became clear that you were most of the time referring to the post-processing method, but I was still bothered throughout the paper by the fact that it could be clearer.

We agree that our use of "model" was too vague and apologise for the confusion this caused.

There are several models used in the work, including the deterministic hydrological model and the post-processing model (which itself comprises the residual error model and the deterministic component). The usage of "model" in the original manuscript was a bit loose and we can see how this caused confusion.

In order to avoid ambiguity/confusion, we now are much more specific regarding which types of models we are referring to. For example, we refer to "streamflow post-processing model" in the abstract (lines 17, 20 and 25).

1.3 For one thing, I would suggest changing the title to indicate that you are interested by comparing two post-processing methods.

Changing the title is a good suggestion.

The original title was *"Seamless streamflow model provides forecasts at all scales from daily to monthly and matches the performance of non-seamless monthly model"*.

We agree the term "streamflow model" could cause confusion. We considered simply replacing "streamflow model" with "streamflow post-processing model" in the original title, but this would make a long title even longer and somewhat obscure. In addition, "post-processing" on its own is itself somewhat ambiguous.

Therefore we decided to make a broader stylistic change to the title, changing it to "Seamless streamflow forecasting at daily to monthly scales: MuTHRE lets you have your cake and eat it too".

This removes the ambiguous term "model" from the title, shortens the title, and conveys the main theme of the paper – i.e. the MuTHRE model provides new capabilities (forecasts all the way from daily to monthly), and still has performance as good as the monthly model.

We have added new text on lines 463-466 to make the connection between the new title and this main theme.

1.4 Second, I would avoid the use of the QPP acronym and instead refer to the "post-processing method" or "post-processing model". I recognize that it would be a bit longer, but in my opinion it would be clearer. I'm one of those persons who don't like acronyms too much...

We agree that the use of acronyms can often make it difficult to follow the text. On the other hand it can also lead to more compact wording.

We have carefully reviewed the use of "QPP" in our paper. The vast majority of times "QPP" is used is in the model name "monthly QPP model". We have elected to continue using this acronym in the model name in order to streamline presentation, and for consistency with figures (where compact model names are required).

Elsewhere in the paper we now use the full term "streamflow post-processing model" (e.g. section heading 3.4.3).

1.5 Section 2 should be named "Post-processing methods" or "Post-processing models", and maybe the titles of the subsections could be adjusted accordingly.

Section 2 was named "Forecasting Models" in the original submission.

We agree that the loose usage of word "model" across the paper created considerable confusion, and appreciate the reviewer suggestion for including "post-processing" in the heading of Section 2.

However, the section content is not limited to post-processing methods – it also includes other components of the forecasting chain, including the rainfall forecasts and deterministic hydrological model. It would be strange and potentially confusing to introduce these components in a section titled "post-processing methods".

We have therefore decided to change the title of Section 2 to "Forecasting methods" to avoid confusion regarding the term "model".

1.6 Finally, I would suggest mentioning GR4J explicitly in section 2.1 and referring to section 3.2 for greater details. In fact, I was initially a bit bothered by the fact that GR4J was described in the case study and not in the "Forecasting models" section, because to me, GR4J is a model and it can be used jointly with meteorological forecasts to produce streamflow forecasts. I think I understand what you wanted to do here: you wanted to avoid drawing too much attention to GR4J because it is not the central element of your study. So in the end, I don't mind too much GR4J being described in section 3.2, but I really think it would help clarify/distinguish things if you could mention it in section 2.1 too.

We originally didn't focus on GR4J in Section 2.1 because the streamflow post-processing models can be used with any deterministic hydrological model, not just GR4J.

However, we do see value in providing some context regarding the term "deterministic hydrological model", and now mention in Section 2.1 that GR4J is the deterministic model used in the case study (lines 106-107).

1.7 This is linked to point 1. When I read the abstract and intro, I had a bit of difficulty understanding what you meant by "designed" for a single time step", because I had, again, a hydrological model in mind. So I was think that, for instance, if GR4J runs at a daily time step, there is nothing stopping you from aggregating it at the monthly time step. So I wasn't completely sure were you were going with that, until I read section 2. It then became very clear that the monthly post-processing method is completely inapplicable at the daily time step and I understood what you meant in the abstract and intro. I think it would be better it could be clearer right from the start, and I think that making it more explicit that you refer to post-processing methods would really help.

Thanks for highlighting this problem. We see that this would have been confusing.

We address this issue by
- Removing the term "streamflow model" from the title, which hopefully prevents an immediate association with a hydrological model.
- Explicitly stating that we compare two post-processing models on line 17 in abstract, to make it clear that we are referring to post-processing models.
- Rewording confusing wording in the abstract (line 14) from "designed" to "developed and applied".
- Elaborating in the introduction that some non-seamless models are developed directly at the monthly scale and do not produce forecasts at the daily scale (lines 46-48).

See also response to comment 1.2 on the use of term "model" across the paper.

1.8 Page 1 line 20: Would it be more accurate to refer to ACCESS-S as an atmospheric model? When I think of a climate model, I think about the ones used for climate change study, that really model the climate over long periods and for which the fluctuations at smaller time steps have no real significance. I know they are basically very similar in their structures, but if I understand correctly, they are not run in the same way at all

The reviewer is correct that "climate model" is not the appropriate term. We now refer to ACCESS-S as a "numerical weather prediction model" in the revised paper.

1.9 I was really thankful for Figure 1. Section 2 was a bit heavy, but this figure is really helpful to understand how everything is put together. Great work.

Thank you. We have further refined Figure 1 and have also re-organised Section 2 to make it easier for the reader to understand both the general structure and specific details of the forecasting methods used in our study.

1.10 Very small point: is it possible that the acronym REM is not defined in the text? Maybe I missed it? I understand it means Residual Error Model but can you please make sure it is defined in the text when first mentioned? Just for clarity.

Good catch. This is now mentioned in the caption to Figure 2.

1.11 Very very small point: Line 111, why chose the subscript "foc"? Why not "fcst" if it is forecasted? I think it would be clearer?

Thanks for the suggestion. We have used the abbreviation "foc" (abbreviated from "forecast") in previous papers on the MuTHRE model (McInerney et al., 2020; McInerney et al., 2021), so prefer to retain this abbreviation for consistency.

1.12 Page 5 line 141: do you mean each individual member or is it really each forecast? I understand you reduce (or "collapse") the forecasts to deterministic ones, so I guess it is really "each forecast", but I'm not sure. Can you maybe clarify? I was a bit lost.

We should have stated that the error model is applied to each individual member (i.e. each ensemble member). We have clarified this in the text (see line 181).

Note for the MuTHRE model we do not collapse the ensemble of forecasts to a deterministic forecast. We see why the previous wording may have suggested otherwise, and believe the revised text will overcome this confusion.

1.13 Page 6 line 167: I like that you use the word "collapse" when referring to the transition from ensemble to deterministic. I think I will adopt this terminology myself in the future.

Nice.

1.14 Section 2.4 is very helpful. Very clear. Thank you.

Thank you!

1.15 Page 10 line 258: I would suggest referring to Schepen et al (2018) as a pre-processing method instead of a post-processing method, especially in the context of your study, since you use the result from this method as an input.

Good point. On line 240 we now say that "the rainfall forecasts are pre-processed using the method of Schepen et al (2018)", and use similar wording elsewhere in the paper.

1.16 I like section 3.4.1. The choice of performance indicators is very well justified and they are clearly appropriate.

Thank you.

1.17 I was surprised that Figure 5 b was not discussed more, especially in relations to the findings in the next page (statistical significance for the different metrics. When looking at Figure 5b, I can't help but think that the monthly post-processing method is not doing a very good job, and this is a bit surprising given that it is trained for monthly forecasts. First, there is an underestimation of streamflow between 2000-2001, and then (2006-2008) a pretty large overestimation. Why is that? Any idea? Then later we learn that, according to the different performance metrics, the differences in performance are actually not often statistically significant. I was surprised by that, because of that figure. I would really appreciate a bit more discussion/explanation of what is happening in 2000-2001 and 2006-2008.

Reviewer 1 is correct that the time series presented in Figure 5b for the monthly QPP model appears inconsistent with the metrics in Figure 6 of the original manuscript.

It turns out that there was a bug in the code used to plot the streamflow time series for the monthly QPP model.

The correct time series in Figure 1 below (which is Figure 6 in the revised paper) shows very little difference between monthly forecasts from the MuTHRE and monthly QPP models. As expected, this is consistent with metrics presented in Figure 7 of the revised paper.

We thank the reviewer for spotting and helping us correct this issue.

[Figure]

*Figure 1: **Revised Figure 6** for the manuscript, showing time series of monthly probabilistic forecasts for Murray River at Biggara.*

1.18 Line 388: I suggest removing the word "actually" to avoid repetition with the previous sentence

Done

**Response to comments from Reviewer 2**

**Summary**

2.1  In this paper, the authors introduce a Multi-Temporal Hydrological Residual Error (MuTHRE) model that enables the production of seamless streamflow forecasts (e.g., daily, weekly, fortnightly, monthly) within the range of 1-30 days. The approach is described and compared against a non-seamless streamflow post-processing (QPP) model implemented by the Australian Bureau of Meteorology's Dynamic Forecasting System. The comparison is performed in 11 Australian catchments in terms of several forecast attributes, and the authors conclude that the MuTHRE model is not only capable of providing good performance for daily streamflow forecasts and cumulative volumes, but also similar performance to that obtained with the non-seamless QPP model for monthly flows.

Overall, this is an interesting manuscript that contributes with encouraging results on the use of seamless streamflow forecasting frameworks. The motivation is clearly stated and the results are nicely presented. There is, nevertheless, a lot of room for improving explanations of the model formulation, streamflow forecast generation, and verification, so that any reader could reproduce the results presented here. There are other minor comments and editorial suggestions that may also help the authors to improve the quality of their manuscript.

We thank Reviewer 2 for their encouraging feedback and detailed review of our paper. In particular, we appreciate their thoughtful suggestions for improving the description of forecasting methods in Section 2, which have made the material easier to follow.

**Main comment:**

2.2  Model description (section 2): I found this section very hard to understand. I think the manuscript would greatly benefit from re-organizing the material and improving definitions and descriptions.

We thank Reviewer 2 for highlighting this issue, and agree that Section 2 in the original paper was difficult to follow.

To address this, we have
- Implemented a more logical introduction to the general forecasting approach in Section 2.1. This includes the use of a new Figure 1 to illustrate the forecasting procedure, and the role of the post-processing model.
- Introduced Figure 2a (schematic of MuTHRE and monthly QPP models) much earlier in Section 2, so that it accompanies the descriptions of the MuTHRE and monthly QPP models in Sections 2.2 and 2.3.
- Introduced a section on general streamflow post-processing model in Section 2.2, and revised Figure 2. This allows us to clearly distinguish the deterministic component and residual error model within the streamflow post-processing models.

We believe that these specific changes, and the other changes suggested by Reviewer 2 below, have made Section 2 much easier to follow.

2.3 For example:

It seems that the two approaches compared here follow the same general model structure (equations 1 and 2). Is that what you mean with "both QPP models"? Can you please be more explicit?

*Yes the two approaches have a similar general structure. We are now more explicit about this in the revised text (lines 95-96), where we state*

*"The forecasting methods investigated in this study share a similar general structure but differ in the streamflow post-processing model."*

2.4 Also, Qt is described as a "probability model for streamflow" (equation 1), and then as a "residual error model" (L103, equation 2) when it is, in reality, the sum of deterministic model output and a residual error term.

*We agree that this original text was confusing and indeed inaccurate.*

*We have revised this description as follows (lines 113-114):*

*"The streamflow post-processing models are constructed using the residual error modelling approach. They comprise a deterministic component and a residual error model."*

2.5 I wonder if you actually need equation (1) in this description.

*We agree and have re-written Section 2.1 without this equation.*

2.6 I think it would be better to have the information presented in L191-214 (differences between MuTHRE and monthly QPP model) right after section 2.1.

*This is a very good suggestion. We now introduce the the MuTHRE and monthly QPP models at the end of Section 2.1 and outline their key features and differences.*

*A major benefit of this reorganisation is that Figure 2 is introduced much earlier. This makes it much easier to follow the descriptions of the MuTHRE and monthly QPP models in Sections 2.2 and 2.3.*

2.7 The authors should consider separating Figure 1 (which is very nice) into two figures: one for model structure (which could include model equations for more direct comparisons between model structures), and another figure for model calibration and forecasting.

*Thanks for this suggestion.*

*Presenting the schematics in Figure 2 (which was Figure 1 in the original paper) in a single figure makes it easier to visually compare the components of the models across different panels.*

*In terms of including equations in the figure, it is difficult to include them without creating clutter and potential confusion. Note that the key differences between the post-processing models – namely aggregating to monthly scale and taking the median – are better described schematically rather than through equations.*

2.8 The meaning of z should be included after presenting equation (2) (perhaps in line 107).

Good suggestion. We have moved the explanation of $z$ earlier, so that it comes directly after equation (2). See line 153.

2.9 Since Xt is also used to describe state variables in the hydrology literature (especially in data assimilation books/papers), I think ut would be more appropriate for meteorological forcings (e.g., Liu and Gupta 2007).

We appreciate this suggestion. Unfortunately notation tends to vary quite a bit across the broad hydrological literature. Since the previous papers on the MuTHRE model (McInerney et al., 2020; McInerney et al., 2021), as well as other papers by our group and other authors, use $\mathbf{x}_t$ for meteorological forcings we prefer to retain this notation for consistency.

2.10 Additionally, in L125 you describe st as a time-varying scaling factor, while the same variable is used to describe hydrological model states in L100.

Good catch. We have changed the symbol for the time-varying scaling factor from $s$ to $\alpha$ .

2.11 L113: I presume that the raw streamflow forecasts do not account for uncertainty in hydrologic model parameters. Can you please clarify?

Correct.

We now
- Mention that the hydrological model has a single set of parameters (which implies no uncertainty in parameters) on line 105, and
- Explicitly state that the we do not consider parametric uncertainty in Section 3.3 when we describe the calibration of hydrological and residual error model parameters (see lines 267-269)

2.12 L135: Is the ensemble size still Nfoc after adding the residual term?

Yes it is. We have clarified this on line 173.

2.13 L141: What do you mean by "individual raw forecast"? Each ensemble member produced with the ensemble of rainfall forecasts?

Yes this is what we meant. We have ensured that consistent terminology is used for ensemble members throughout the paper.

2.14 L148: how is m* determined?

It is computed as the mean of the residual $\boldsymbol{\eta}$ after the seasonality and dynamic bias terms are removed. We now mention this on lines 187-188.

2.15 Since the paper should be self-contained, additional information on the calibration procedures referred to in L162 and L190 should be provided (what are the calibration period, objective functions, and optimization algorithms?). A couple of sentences should suffice.

Good suggestion. We now describe how the parameters in the streamflow post-processing models are estimated in Sections 2.3.4 and 2.4.4 (see lines 202-204 and 225).

We note that Section 3.3 provides details of cross-validation procedure (including calibration periods), as well as the objective function and optimization algorithm for calibrating the hydrological model.

2.16 L167: "and then collapsed to a deterministic forecast by taking the median". Is this current operational practice?

This step is performed in the Bureau of Meteorology's dynamic forecasting system (see Section 2.3.2 in Woldemeskel et al., 2018).

2.17 L111, L112, L135, L168, L169, and elsewhere: is "replicate" the same as "ensemble member"?

Yes "replicate" is the same as "ensemble member". We have ensured "ensemble member" is used throughout paper.

**Additional minor comments**

2.18 L33-35: It makes more sense to me to describe common practice before referring to the need for seamless forecasts. Also, it would be worth highlighting that non-seamless forecasting efforts have been (and are being) conducted in South America (e.g., Souza Filho and Lall 2003; Mendoza et al. 2014), Europe (e.g., Ionita et al. 2008; Hidalgo-Muñoz et al. 2015), Asia (e.g., Pal et al. 2013) and everywhere else around the world, with appropriate citations.

Thanks for this suggestion.

We originally referred to "seamless forecasts" first, because it made it easier to define a "non-seamless forecast". However, it does make sense to start with common practice, and have implemented this suggested change. We appreciate the references for other non-seamless forecasts around the world, and have included some of these in the introduction (line 33).

2.19 L37: "This is the focus of our study". This reads out of place here. I recommend deleting this sentence or moving it toward the end of the introduction.

We agree and have removed this sentence.

2.20 Figure 2: How many values are contained in each boxplot? One per basin?

There are 11 values in each boxplot, corresponding to the 11 catchments. We have clarified this in the caption.

2.21 Since you have only 11 catchments, I think it would be better to show one line per basin.

Thanks for the suggestion. We did try this, but preferred the boxplots because
- They are cleaner (fewer lines), which makes it easier to see variability between months and years
- All catchments show roughly the same patterns for both monthly and annual variability, so there is not much to be gained from showing each catchment

2.22 Further, it would be informative for readers to have a table with the name of the station, basin-averaged elevation, area, mean annual runoff, mean annual precipitation, mean annual temperature, annual runoff ratio, and aridity index.

Good suggestion. The revised manuscript includes a new Table 1 with relevant catchment information.

2.23 L273: Are you working with calendar years or water years?

Calendar years. We have clarified this in the text (line 261).

2.24 Are daily forecasts produced each day in year j with MuTHRE, or only at the beginning of each month?

Forecasts for the MuTHRE model are produced at the start of the month only. We have clarified this on line 272-273.

2.25 What is the final ensemble size of your forecasts?

The size of the post-processed streamflow ensembles is 100, which is the same as size of the forecast rainfall ensembles. We have clarified this in the text (line 271).

2.26 L274-275: The problem of hydrologic memory in Australian catchments and its implications for cross-validation has been previously documented (e.g., Robertson et al. 2013; Pokhrel et al. 2013). I recommend the authors read and cite these papers here. The following blog article is also relevant: https://hepex.inrae.fr/how-good-is-my-forecasting-method-some-thoughts-on-forecast-evaluation-using-cross-validation-based-on-australian-experiences/

Thank you for these references. We have added a citation regarding the importance of dealing with hydrologic memory in the cross-validation procedure (see line 263).

2.27 L292: Perhaps it would be better to replace the word "uncertainty" with "spread".

Good suggestion. We have made this change (see line 282).

2.28 Also, it would be informative to state that sharpness is a forecast attribute only (i.e., it does not depend on the observations).

Good suggestion. We have added this to lines 285-286.

2.29 L296: Since CRPS measures the difference between forecast and observation CDFs, it would be better to refer to "probability forecast errors" instead of "combined performance".

We describe CRPS as a metric for "combined" performance because, as shown in Hersbach (2000), the CRPS can be decomposed into terms representing individual performance aspects, namely reliability, and uncertainty/resolution (related to sharpness). We agree this may not have been obvious to a general reader and have now clarified this in the text (see lines 292-293).

Note that the CRPS serving as a combined performance metric is very relevant to our study because the other three metrics – namely reliability, sharpness and bias - focus on fundamentally more specific performance characteristics. If we do not highlight this, readers may form the erroneous impression that we have four independent performance metrics.

In order to convey that CRPS represents "probability forecast errors", we now mention on lines 290-291 that

"the CRPS is defined as the sum of squared differences between forecast cumulative distribution function (CDF) and the empirical CDF of the observation"

2.30 Figure 4: How are confidence limits generated? Do you compute the metric merging forecasts from all basins? Please clarify these points in the manuscript.

Throughout the paper metrics are computed separately for each catchment and the confidence limits in figures show the minimum and maximum values (i.e. the range) over the 11 catchments. This is now described in the caption in Figure 5 (old Figure 4), and other captions.

2.31 L368-370: You mention that reliability results are similar, although the boxplots look different. I recommend applying a statistical test to determine whether the distributions of these metrics are significantly different.

We do use a statistical test to evaluate difference in distributions of metric values. Specifically, we use "practical significance testing" (described in Section 3.4.3) to determine whether differences between models are not just statistically significant, but are statistically significantly larger than some pre-defined practically relevant margin (chosen as 20% of metric value).

Although the distributions in Figure 7a appear different, these differences are not practically significant based on the criteria defined in Section 3.4.3.

2.32 L370 and elsewhere: "practically significant" or "significant". Are the authors referring to a statistically significant result? If not, I suggest re-wording or deleting the word 'significant'.

Following on from the earlier point 2.31, the term "practically significant" refers to cases where the difference in metric values are statistically significantly larger than a pre-defined margin.

We realise that we did not explicitly define the term "practically significant differences" in Section 3.4.3 of the original manuscript, which may have led to confusion.

We have made the following changes in order to make it clearer what "practically significant" refers to
1. We have added "practically significant differences" to the heading for Section 3.4.3
2. We now refer to Section 3.4.3 when we first mention "practically significant" in the Results section (see line 370).

2.33 Figure 6: I think you should say "overall monthly performance" in the caption, and perhaps remind readers here what "overall" means.

This is a good suggestion. The caption for Figure 7 (old Figure 6) now reads

*"Overall performance (all months and years, …"*

2.34 Are you grouping the results of all basins? In the left panels, how many points are contained in each boxplot?

Yes we are showing the results from all catchments in the boxplots. Each boxplot represents the distribution of metric values from the 11 catchments. We have clarified this in the caption to Figure 7.

2.35 In the center and right panels, how are the confidence limits computed?

The limits represent the minimum and maximum values over the 11 catchments (see comment 2.30). This is now described in the caption of Figure 7.

2.36 L421: Shall we expect persistence in rainfall, given the chaotic nature of the atmosphere?

Good question.

In this sentence we are referring to the day-to-day persistence in rainfall, which will be important for reliable monthly rainfall forecasts. We are not referring to longer-term persistence, which will be much smaller due to the chaotic nature of the atmosphere.

We have clarified that we are referring to day-to-day persistence in the revised paper (see line 422).

2.37 L426: I encourage the authors to replace the last sentence of this paragraph (which reads a lot like "propaganda") with a more quantitative statement regarding the performance of MuTHRE.

We have reworded this sentence (lines 427-428) to

*"The performance at the monthly scale is particularly encouraging given monthly data is not used in calibration."*

which we believe is factual and focuses on the key point we are trying to make.

2.38 L467: "This was not feasible in this study". If you cannot provide an explanation on why was not feasible, I suggest deleting this sentence.

We agree and have removed this sentence.

2.39 L479: "High-quality forecasts". Note that the quality depends a lot on the forecast attributes you are analyzing. I think it would be good to provide a brief discussion (maybe in section 5) about tradeoffs between the metrics included here (e.g., how your forecast system can improve reliability at the cost of losing sharpness), and what makes a forecast "good" or "high-quality".

We agree that our classification of "high-quality forecasts" is dependent on the forecast attributes/metrics considered in this study.

We now make this point by changing this phrase on lines 491-492 to

*"high quality forecasts (based on the metrics considered in this study)"*

Not that the results of our case study show that differences in reliability for the two models are not practically significant. As such, there is no (practically relevant) trade-off between the reliability and sharpness of forecasts to discuss.

**Suggested edits**

2.40 L51: 'Hydro-electric' -> 'hydropower'.

L70: 'drop in' -> 'loss of'.

L312: 'which' -> 'who'.

L377: 'in 1 month (September)' -> 'in September'.

L380: delete 'similar/better performance in all months, with practically'.

L382: delete 'similar/better performance in 19 out of 22 years, with practical'.

L451: 'Simplifies' -> 'A simplified'.

L473: 'to forecasts' -> 'compared to forecasts'.

Thank you for these suggested edits. We have implemented these changes.

**References**

Hersbach, H. 2000. Decomposition of the Continuous Ranked Probability Score for Ensemble Prediction Systems. *Weather and Forecasting,* 15**,** 559-570.

McInerney, D., Thyer, M., Kavetski, D., Laugesen, R., Tuteja, N. & Kuczera, G. 2020. Multi-temporal hydrological residual error modelling for seamless sub-seasonal streamflow forecasting. *Water Resources Research,* 56**,** e2019WR026979.

McInerney, D., Thyer, M., Kavetski, D., Laugesen, R., Woldemeskel, F., Tuteja, N. & Kuczera, G. 2021. Improving the Reliability of Sub-Seasonal Forecasts of High and Low Flows by Using a Flow-Dependent Nonparametric Model. *Water Resources Research,* 57**,** e2020WR029317.

Woldemeskel, F., McInerney, D., Lerat, J., Thyer, M., Kavetski, D., Shin, D., Tuteja, N. & Kuczera, G. 2018. Evaluating residual error approaches for post-processing monthly and seasonal streamflow forecasts. *Hydrol. Earth Syst. Sci. Discuss.,* 2018**,** 1-40.

---

## Referee Report (RR1)

**Reviewer 1 comments on manuscript hess-2021-589**
**« Seamless streamflow forecasting at daily to monthly scales: MuTHREE lets you have your cake and it too »**

I am very happy with the authors' response and with the revised version of the manuscript. I do not have any further comments and I think it should now be published as is.